# Titanocene Selenide Sulfides Revisited: Formation, Stabilities, and NMR Spectroscopic Properties

**DOI:** 10.3390/molecules24020319

**Published:** 2019-01-16

**Authors:** Heli Laasonen, Johanna Ikäheimonen, Mikko Suomela, J. Mikko Rautiainen, Risto S. Laitinen

**Affiliations:** 1Laboratory of Inorganic Chemistry, Environmental and Chemical Engineering, University of Oulu, P.O. Box 3000, 90014 Oulu, Finland; heli.laasonen@valvira.fi (H.L.); johanna.ikaheimonen@outokumpu.com (J.I.); MikkoSuomela@eurofins.fi (M.S.); 2Department of Health, Legal Rights and Technologies, National Supervisory Authority for Welfare and Health (Valvira), P.O. Box 210, 00281 Helsinki, Finland; 3Outokumpu Stainless Oy, Terästie 1, 95490 Tornio, Finland; 4Eurofins Nab Labs Oy, Industry Services, Nuottasaarentie 17, 90400 Oulu, Finland; 5Department of Chemistry, Nanoscience Centre, University of Jyväskylä, P.O. Box 35, 40014 Jyväskylä, Finland; j.mikko.rautiainen@jyu.fi

**Keywords:** titanocene selenide sulfides, ^77^Se-NMR spectroscopy, ^13^C-NMR spectroscopy, crystal structures, DLPNO-CCSD(T) calculations

## Abstract

[TiCp_2_S_5_] (phase ***A***), [TiCp_2_Se_5_] (phase ***F***), and five solid solutions of mixed titanocene selenide sulfides [TiCp_2_Se*_x_*S_5−*x*_] (Cp = C_5_H_5_^−^) with the initial Se:S ranging from 1:4 to 4:1 (phases ***B***–***E***) were prepared by reduction of elemental sulfur or selenium or their mixtures by lithium triethylhydridoborate in thf followed by the treatment with titanocene dichloride [TiCp_2_Cl_2_]. Their ^77^Se and ^13^C NMR spectra were recorded from the CS_2_ solution. The definite assignment of the ^77^Se NMR spectra was based on the PBE0/def2-TZVPP calculations of the ^77^Se chemical shifts and is supported by ^13^C NMR spectra of the samples. The following complexes in varying ratios were identified in the CS_2_ solutions of the phases ***B***–***E***: [TiCp_2_Se_5_] (**5_1_**), [TiCp_2_Se_4_S] (**4_1_**), [TiCp_2_Se_3_S_2_] (**3_1_**), [TiCp_2_SSe_3_S] (**3_6_**), [TiCp_2_SSe_2_S_2_] (**2_5_**), [TiCp_2_SSeS_3_] (**1_2_**), and [TiCp_2_S_5_] (**0_1_**). The disorder scheme in the chalcogen atom positions of the phases ***B***–***E*** observed upon crystal structure determinations is consistent with the spectral assignment. The enthalpies of formation calculated for all twenty [TiCp_2_Se*_x_*S_5−*x*_] (*x* = 0–5) at DLPNO-CCSD(T)/CBS level including corrections for core-valence correlation and scalar relativistic, as well as spin-orbit coupling contributions indicated that within a given chemical composition, the isomers of most favourable enthalpy of formation were those, which were observed by ^77^Se and ^13^C NMR spectroscopy.

## 1. Introduction

Titanocene pentasulfide [bis(cyclopentadienyl)pentasulfidotitanium] and titanocene penta-selenide [bis(cyclopentadienyl)pentaselenidotitanium] {the general formula [TiCp_2_E_5_] (Cp = C_5_H_5_^−^ or its alkyl-substituted derivative; E = S, Se)}, as well as the related dinuclear complexes [TiCp_2_(μ-E*_n_*)_2_TiCp_2_] (*n* = 2 or 3, the latter in case of sulfur) have long been known as convenient chalcogen atom transfer reagents (for some selected reviews see [1,2,3,4,5,6,7,8]). These reagents are particularly useful in the preparation of homocyclic sulfur and selenium ring molecules as well as individual heterocyclic selenium sulfides, which are generally formed only in complicated molecular mixtures.

Crystal structures have been reported for mononuclear titanocene pentasulfide [9,10,11,12], pentaselenide [13,14], and mixed titanocene selenide sulfides [15,16,17], as well as the related dinuclear complexes [18,19,20,21]. Their NMR spectroscopic properties are also well known [15,16,18,19,20,21,22,23,24,25,26,27]. This information was utilized in the preliminary identification of individual molecular components in the mixtures of titanocene selenide sulfides [TiCp_2_Se*_x_*S_5−*x*_] [15]. The assignment of the ^77^Se NMR spectra of mixtures containing different initial molar ratios of sulfur and selenium was based on the consideration of the spectroscopic data of [TiCp_2_Se_5_] [25], on the constant intensities between some resonances, as the Se:S ratio is varied, and on the trends in the ^77^Se chemical shifts [28,29,30]. This semi-quantitative analysis has then been utilized in resolving the disorder in the chalcogen atom positions in the crystalline [TiCp_2_Se*_x_*S_5−*x*_] phase, which was prepared using the initial Se:S molar ratio of 3:2 [15]. The composition was further verified by treating the [TiCp_2_Se*_x_*S_5−*x*_] mixture with Se_2_Cl_2_ or S_2_Cl_2_ and determining the composition of the two product mixtures using ^77^Se NMR spectroscopy [31].

In this contribution, we extend these earlier studies by considering the stabilities of different [TiCp_2_Se*_x_*S_5−*x*_] complexes using high-level DFT and domain based local pair natural orbital coupled cluster [DLPNO-CCSD(T)] computations. We also verify the earlier spectroscopic assignments by computing the ^77^Se shielding tensors and chemical shifts utilizing the methodology, which has proven suitable for molecular selenium species [32]. For understanding the trends in the formation of individual complexes, we have considered six different syntheses by varying the initial Se:S molar ratio in the preparations. The designation of the product phases are as follows: Phase ***A*** (only sulfur), phase ***B*** (Se:S = 1:4), phase ***C*** (Se:S = 2:3), phase ***D*** (Se:S = 3:2), phase ***E*** (Se:S = 4:1), and phase ***F*** (only selenium). Since the crystal structures of phase ***A*** {[TiCp_2_S_5_]} [9,10,11,12], phase ***F*** {[TiCp_2_Se_5_]} [13,14], and phase ***D*** [15] have been determined previously, in this contribution we have determined only the crystal structures of Phases ***B***, ***C***, and ***E***. The ^77^Se NMR spectra of phases ***C***, ***D***, and ***F*** are also known [15,24]. In this work, we augment this information by those of ***B*** and ***E***. The ^13^C NMR spectra of all five mixed phases ***B***–***E*** as well as those of [TiCp_2_S_5_] (***A***) and [TiCp_2_Se_5_] (***F***) are reported. The main objective of the present work is to get further information of the composition of the crystalline solid solutions of the [TiCp_2_Se*_x_*S_5−*x*_] phases.

## 2. Results and Discussion

### 2.1. Crystal Structures of Phases **B**, **C**, and **E**

It has been well-established that the reduction of sulfur-selenium mixtures by Li[AlEt_3_H] followed by the reaction with [TiCp_2_Cl_2_] forms a mixture of [TiCp_2_Se*_x_*S_5−*x*_] complexes [15] in an analogous manner to the formation of [TiCp_2_S_5_] and [TiCp_2_Se_5_] from sulfur and selenium [1,2,3,4,5,6,7,8]. There are 20 possible complexes in the [TiCp_2_Se*_x_*S_5−*x*_] (*x* = 0–5) series, which are shown in Figure 1 together with their abbreviated designations. Upon crystallization, they form solid solutions.

The crystal structures of [TiCp_2_S_5_] (Phase ***A***) [9,10,11], [TiCp_2_Se_5_] (Phase ***F***) [13,14], and Phase ***D*** [15] are well-known. The crystal structures of Phases ***B***, ***C***, and ***E*** have been determined in this work. All phases ***A***–***F*** show similar structures of the bidentate anionic chelating E_5_^2−^ ligand (E = S, Se) coordinating to a titanium atom of the bis(cyclopentadienyl)titanium fragment, as shown in Figure 2.

Four different isomorphic series have been observed for crystalline phases ***A***–***F***. [TiCp_2_S_5_] (Phase ***A***) shows two different monoclinic polymorphs [9,10,11]. The mixed sulfur-selenium phases ***B***–***D*** are mutually isomorphic and are also isomorphic with [ZrCp_2_Se_5_] [14]. [TiCp_2_Se_5_] (phase ***F***) is triclinic and is isomorphic with phase ***E*** [13,14].

With the exception of atoms E2, E3, and E4 in phase ***E***, all chalcogen atom positions in phases ***B***–***E*** are disordered with sulfur and selenium distributed randomly in the atomic sites. The site occupancy factors of the chalcogen atoms in Phases ***B***–***E*** are shown in the Appendix A (Appendix A). Because of the disorder, the bond distances between the chalcogen atoms only reflect the average disordered composition of the atomic sites and therefore carry no accurate physical significance. These interatomic distances are also listed in the Appendix A (Appendix A).

### 2.2. ^77^Se and ^13^C-NMR Spectra of the [TiCp_2_Se_x_S_5−x_] (x = 0–5) Phases **A**–**F**

A typical ^77^Se NMR spectrum of the [TiCp_2_Se*_x_*S_5−*x*_] mixture has been recorded previously for the Phase ***D*** [15]. The resonances were tentatively assigned to [TiCp_2_Se_5_] (**5_1_**), [TiCp_2_Se_4_S] (**4_1_**), [TiCp_2_Se_3_S_2_] (**3_1_**), [TiCp_2_SSe_3_S] (**3_6_**), [TiCp_2_SSe_2_S_2_] (**2_5_**), and [TiCp_2_SSeS_3_] (**1_2_**) [see Figure 3a]. The spectrum of phase ***E*** is somewhat simpler but shows only resonances, which are also observed for other phases [see Figure 3b]. The current study indicates that no new resonances were observed in the complete range of the Se:S ratio from 1:4 to 4:1. Only the relative intensities of the resonances varied.

It was observed that by varying the Se:S ratio of the chalcogen reagent the relative intensities of some groups of resonances remained constant, while the relative intensities between the groups varied. The groups exhibiting constant intensity ratios are shown in Table 1 and formed the basis for the initial assignment reported earlier [15].

The assignment was verified by PBE0/def2-TZVPP calculations of nuclear magnetic shielding tensors using methodology described previously [32]. The computed ^77^Se chemical shifts of all nineteen selenium-containing [TiCp_2_Se*_x_*S_5−*x*_] complexes are presented in Table 1 and compared with those of the experimental resonances. It can be seen from Table 1 that the current calculations agree very well with the previous tentative assignment [15] of the ^77^Se resonances. The agreement between the computed and observed ^77^Se chemical shifts is excellent and provides the best fit between the experiment and theory. It was reported earlier [15] that it is not possible to assign unambiguously the ^77^Se NMR resonance at 936 ppm either to [TiCp_2_SSeS_3_] (**1_2_**) or [TiCp_2_S_2_SeS_2_] (**1_3_**). The current computations, however, indicate that **1_2_** is a more likely candidate.

Since it has been deduced earlier from the crystallographic disorder scheme that the Phase ***D*** must also contain [TiCp_2_S_5_] (**0_1_**) [15], which cannot be detected by ^77^Se NMR spectroscopy, the ^13^C NMR spectra of all phases ***A***–***F*** were recorded in this contribution. All complexes present in the mixtures show ^13^C NMR resonances due to the carbon atoms in the cyclic η^5^-C_5_H_5_ ligand. ^1^H NMR spectroscopy has shown that in solution at room temperature, the ligand is rotationally fluxional [23,24]. Since it is also known that conformational chair-chair inversion of the hexaatomic TiE_5_ chelate ring does not take place under ambient conditions [23,24], two resonances are observed in ^1^H and ^13^C NMR spectra for each [TiCp_2_Se*_x_*S_5−*x*_] complex (for a typical ^13^C NMR spectrum, see Figure 4).

The assignment in the ^13^C NMR spectrum shown in Figure 4 was based on the comparison of relative intensities of the pairs ^13^C resonances of equal intensity with those obtained from the corresponding ^77^Se spectrum. The possibility to prepare and record the ^13^C NMR spectra of pure [TiCp_2_S_5_] (**0_1_**) and [TiCp_2_Se_5_] (**5_1_**) served to verify the assignments. Further support to the assignments in Figure 4 is obtained by considering the trends in the chemical shifts. In case of [TiCp_2_Se_5_] (**5_1_**), which contains two Ti–Se bonds, the average of the two carbon shifts is 111.5 ppm. [TiCp_2_Se_4_S] (**4_1_**) and [TiCp_2_Se_3_S_2_] (**3_1_**) have one Ti–S bond and one Ti–Se bond. Their average ^13^C chemical shifts span a narrow range of 112.1–112.3 ppm. Other complexes, which contain two Ti–S bonds, i.e., [TiCp_2_SSe_3_S] (**3_6_**), [TiCp_2_SSe_2_S_2_] (**2_5_**), [TiCp_2_SSeS_3_] (**1_2_**), and [TiCp_2_S_5_] (**0_1_**) show their average chemical shifts in the range 112.5–113.0 ppm. The trend of decreasing shielding in the carbon atoms is to be expected, since the electronegativity of sulfur is slightly higher than that of selenium.

Abel et al. [24] have inferred that the ^13^C resonance of fluxional cyclopentadienyl ring in the axial position is more shielded than that in the equatorial position. The trend in the difference between the two chemical shifts is very informative in the [TiCp_2_Se*_x_*S_5−*x*_] series, as shown in Figure 5. It can be seen that the numerical values of the differences can be classified in three distinct groups. The interpretation of the trend can be explained as follows:

The twenty complexes can have their five chalcogen atom positions occupied either by sulfur or selenium resulting in a varying number of homo- and heteronuclear chalcogen-chalcogen bonds. [TiCp_2_S_5_] (**0_1_**) and [TiCp_2_Se_5_] (**5_1_**) contain no heteronuclear S–Se bonds and both show a large difference of 0.94 and 0.88 ppm between the chemical shifts of the fluxional equatorial and axial C_5_H_5_^−^ ligands. [TiCp_2_Se_4_S] (**4_1_**) and [TiCp_2_Se_3_S_2_] (**3_1_**) both contain one heteronuclear S–Se bond either in the position marked by *r*_1_ or *r*_2_ (see Figure 5) and show a very small chemical difference of 0.17–0.18 ppm. Other [TiCp_2_Se*_x_*S_5−*x*_] complexes show a total of two Se–S bonds either in the position *r*_1_ or *r*_2_. They show chemical differences in the range 0.47–0.65 ppm.

### 2.3. Composition of [TiCp_2_Se_x_S_5−x_] Phases **B**–**E**

The approximate compositions of the phases ***B***–***E***, which have been determined by considering the intensities of the ^77^Se and ^13^C resonances, are shown in Table 2.

The composition of the solid phases ***B***–***E*** crystallized from the reaction mixtures can be estimated considering the disorder scheme in the crystal structures and assuming that the solid solutions contain only complexes, which have been detected in solution (for details, see Appendix A). These semiquantitative analytical results are presented in Table 3.

Expectedly, all solid solutions are richer in selenium than the solutions or the initial reagents of the elemental selenium-sulfur mixtures. This is due to the decreasing solubility of the complexes as the selenium-content increases. It can be inferred, however, that disorder scheme observed in the crystal structures is consistent with the complexes, which are formed in the reactions. The justification for these conclusions is presented in Appendix A.

### 2.4. Relative Stabilities of Individual [TiCp_2_Se_x_S_5−x_] (x = 0–5) Complexes

Enthalpies of formation for individual [TiCp_2_Se_x_S_5−*x*_] species have been predicted using DLNPO-CCSD(T)/CBS energies. Test computations on reference molecules Se_2_ and S_2_ showed that the present calculations are capable of predicting the enthalpies of formation of chalcogen species with very good accuracy {c.f. Δ_f_*H*(Se_2_) calc. +146.8 vs. exptl. +144.9 ± 1.1 kJ mol^−1^ [33] and Δ_f_*H*(S_2_) calc. +130.6 vs. exptl. +128.6 ± 0.3 kJ mol^−1^ [34]}. The calculated formation enthalpies of [TiCp_2_Se_x_S_5−*x*_] species shown in Figure 6 and in Appendix A in Appendix A indicate that the formation reactions of all complexes are exothermic in the gas phase. The calculated Δ_f_*H* values span a relatively small range (<30 kJ mol^−1^), as would be expected for the species forming an equilibrium in solution. [TiCp_2_S_5_] (**0_1_**) is predicted to be the most stable of [TiCp_2_Se_x_S_5−*x*_] species. The sulfur-rich [TiCp_2_Se_x_S_5−*x*_] complexes are in general more stable than the selenium-rich species. It can be seen from Figure 6 that the formation enthalpies can be classified in three distinct, but slightly overlapping groups based on the number of Ti–S and Ti–Se bonds.

Δ_f_*H*_nonrel._ values, which have been calculated without scalar relativistic correction Δ*E* (C + R) and atomic spin-orbit Δ*E* (SO) correction, have been included in Appendix A (see Appendix A) for comparison. They show that the inclusion of core-valence correlation and scalar relativistic correction to energies stabilizes the calculated structures significantly compared to free elements [35,36]. Addition of the spin-orbit corrections has an opposite effect on the calculated enthalpies especially in the case of selenium-rich species and changes the stability order in the [TiCp_2_Se_x_S_5−*x*_] series to favor sulfur-rich species.

Each observed [TiCp_2_Se_x_S_5−*x*_] species in the CS_2_ solutions correspond to the most stable structures in the series of isomers with the same composition (see Figure 7). The only exception is the complex **1_2_** which lies 1.1 kJ mol^−1^ higher in energy than the lowest energy isomer **1_3_**. It should be noted that the solvent effects were not considered in calculated Δ_f_*H* values. They do not have a significant effect on the relative stabilities.

The current computations show that isomeric structures with maximal number of homopolar chalcogen bonds are stabilized over structures with chalcogen heteroatom bonds (S–Se). This has been observed to be the case also for heterocyclic Se*_x_*S_8−*x*_ molecules [28,30]. This is also consistent with the observation that the formation of SeS(g) from S_2_(g) and Se_2_(g) is slightly endothermic (5.2 kJ mol^−1^) [32]. The structures with S–Ti bonds are favoured over Se–Ti bonds.

## 3. Experimental

### 3.1. Preparation of [TiCp_2_Se_x_S_5−x_]

All reactions and manipulations of air- and moisture-sensitive materials were carried out under an argon atmosphere by using a standard drybox or Schlenk techniques. Tetrahydrofuran (LabScan, Bangkok, Thailand) was dried before use by distillation over Na/benzophenone in a nitrogen atmosphere, and carbon disulfide (Thermo Fisher Scientific, Waltham, MA, USA) was distilled over P_4_O_10_ also under a nitrogen atmosphere. Sulfur (Merck, Darmstadt, Germany), selenium (Merck), and [TiCp_2_Cl_2_] (Cp = C_5_H_5_^−^, Sigma-Aldrich, Darmstadt, Germany) were used as provided. [TiCp_2_S_5_], [TiCp_2_Se_5_] and all [TiCp_2_Se*_x_*S_5−*x*_] (*x* = 1–4) mixtures were prepared as described previously modifying the method by Gladysz et al. [15,37] (for synthetic details, see Appendix A). After the reaction, the solutions were filtered, the solvent was evaporated, and the solid material was extracted by carbon disulfide. The ^77^Se- and ^13^C-NMR spectra were recorded from the thus formed CS_2_ solution. The crystals obtained upon recrystallization of these CS_2_ solutions were involved in the crystal structure determinations. The designation of the final crystalline products was based on the initial Se:S molar ratio of the elemental chalcogen mixture reagent, as follows: Phase ***B*** (Se:S = 1:4), phase ***C*** (Se:S = 2:3), phase ***D*** (Se:S = 3:2), and phase ***E*** (Se:S = 4:1).

### 3.2. NMR Spectroscopy

The ^77^Se and ^13^C NMR spectra were recorded in the CS_2_ solution on a DPX-400 spectrometer (Bruker, Karlsruhe, Germany) operating at 76.31 and 100.61 MHz, respectively. The spectra were recorded unlocked. Typical respective spectral widths for ^77^Se and ^13^C were 76,000, and 30,000 kHz, and the respective pulse widths were 6.7 and 4.00 µs. The pulse delay for selenium was 0.43 s and for carbon 0.81 s. The ^77^Se-NMR spectra were referenced externally to a saturated aqueous solution of selenium dioxide. The chemical shifts are reported relative to neat Me_2_Se [δ(Me_2_Se) = δ(SeO_2_) + 1302.6] [38]. In case of ^13^C spectra, the chemical shifts were referenced and reported relative to TMS.

### 3.3. X-ray Crystallography

Diffraction data for crystal phases ***B***, ***C***, and ***E*** were collected on a Nonius Kappa CCD diffractometer (Bruker, Karlsruhe, Germany) at 120 K using graphite-monochromated Mo K_α_ radiation (λ = 0.71073 Å; 55 kV, 25 mA). Crystal data and the details of structure determinations are given in Table 4. All structures were solved by direct methods using SHELXS-2016 and refined using SHELXL-2016 [39,40]. After the full-matrix least-squares refinement of the non-hydrogen atoms with anisotropic thermal parameters, the hydrogen atoms were placed in calculated positions in the cyclopentadienyl groups (C–H = 0.95 Å). The isotropic thermal parameters of the hydrogen atoms were fixed at 1.2 times to that of the corresponding carbon or nitrogen. The scattering factors for the neutral atoms were those incorporated with the program.

All chalcogen atom positions in phases ***B*** and ***C*** were disordered with sulfur and selenium statistically distributed over the atomic sites. In case of phase ***E*** the chalcogen atoms 1 and 5 were also disordered, while those of 2–4 were only occupied by selenium. The following constraints were applied due to the correlation between the thermal parameters and the occupation factor:sof(Se*_i_*) + sof(S*_i_*) = 1(1)
*U*(Se*_i_*) = *U*(S*_i_*)(2)
where sof(Se*_i_*), sof(S*_i_*), *U*(Se*_i_*), and *U*(S*_i_*) are the occupation factors and isotropic thermal parameters or the main diagonal parameters in the anisotropic thermal parameter tensor of selenium and sulfur atoms at the *i*^th^ atomic position. Sulfur and selenium atoms of the disordered pairs were also constrained in the same atomic positions. Information on physically meaningful bond lengths and bond angles was thereby lost. The method, however, enables the reliable refinement of the occupation factors of selenium and sulfur at the disordered chalcogen atom sites and thus serves in the identification of the molecular species.

The X-ray data can be obtained free of charge via www.ccdc.cam.ac.uk/conts/retrieving.html or from the Cambridge Crystallographic Data centre, 12 Union Road, Cambridge CB2 1EZ, UK; fax (+44) 1223-336-033; e-mail: deposit@ccdc.cam.ac.uk (for CCDC registry numbers, see Table 4).

## 4. Computational Details

All structures were optimized using PBE0 hybrid DFT [41,42,43] with def2-TZVPP basis sets [44] on all atoms and employing empirical D3BJ correction [45] to treat dispersion forces. Nuclear magnetic shielding tensors were calculated on optimized stationary points with GIAO method [46,47,48,49] as implemented in Gaussian 09 [50] that was used for all DFT calculations. ^77^Se chemical shifts were determined from calculated nuclear shielding tensors using a previously described calibration [32].

Enthalpies of formation were determined using electronic energies calculated with DLPNO-CCSD(T) method [51,52,53,54] implemented in ORCA 4.0 program suite [55] that has been recently shown to provide accurate enthalpy estimations with moderate computational cost when used together with thermal energy corrections from DFT calculations [56,57]. TightPNO setting, that controls the cut-off parameters for the treatment of the domain based localized pair natural orbitals [58], and VeryTightSCF (energy change 1 × 10^−9^ au) option to achieve better converged reference wavefunctions were used in the energy calculations. DLPNO-CCSD(T) energies calculated with def2-XZVPP (X = T, Q) basis sets [41], and corresponding auxiliary basis sets def2-XZVPP/C [59], were extrapolated to complete basis set (CBS) limit using separate extrapolations for reference Hartree-Fock energy [60] and correlation energy [61].
(3)$$E_{\mathrm{HF}}^{X}=E_{\mathrm{HF}}^{\infty }+{\mathrm{Ae}}^{-\alpha \sqrt{X}}$$
(4)$$E_{\mathrm{corr}.}^{X}=E_{\mathrm{corr}.}^{\infty }+{\mathrm{BX}}^{-\beta }$$
where α (7.88) and β (2.97) are optimized coefficients for the basis set couple taken from Neese and Valeev [62] and A and B are parameters to be obtained by combining the results of the two basis set levels.

Neglect of core-valence correlation and scalar relativistic effects have been shown to be potential sources of significant error in relative energies [35]. However, calculation of core-valence correlation and scalar relativistic effects are likely to be the most time-consuming step in the calculations if treated at the same level as the valence correlation calculations [63,64]. To account for core-valence correlation and scalar relativistic contributions to energies while conserving computational resources we adopted a similar approach used by Chan and Radom in W1X-1 composite method [64] and calculated the contributions at DLPNO-MP2/cc-pwCVTZ level [36,65,66,67,68] as differences between frozen-core and all-electron-DKH energies and added the contribution as correction Δ*E* (C + R) term to total energies. Spin-orbit coupling corrections Δ*E* (SO) were applied to atomic reference energies using weighted *J*-averaged values derived from experimental data in Moore’s tables [69].

Enthalpies of formation were calculated from atomization enthalpies using reference values taken from tables published by the committee on Data of the International Council for Science (CODATA) [34] for other elements and from the work of Drowart and Smoes [33] for selenium.

## 5. Conclusions

It has long been known that the reduction of elemental sulfur or selenium by lithium triethylhydridoborate in THF followed by the treatment with titanocene dichloride [TiCp_2_Cl_2_] (Cp = C_5_H_5_^−^ or its alkyl-substituted derivative) affords [TiCp_2_S_5_] or [TiCp_2_Se_5_], respectively [9,10,11,12,13,14]. When mixtures of elemental sulfur and selenium are used as the reagent in the reaction, mixtures of [TiCp_2_Se*_x_*S_5−*x*_] were formed [15]. Tentative identification of the molecular species formed in the reactions with the initial Se:S molar ratio of 2:3 and 3:2 indicated the formation of [TiCp_2_Se_5_] (**5_1_**), [TiCp_2_Se_4_S] (**4_1_**), [TiCp_2_Se_3_S_2_] (**3_1_**), [TiCp_2_SSe_3_S] (**3_6_**), [TiCp_2_SSe_2_S_2_] (**2_5_**), and [TiCp_2_SSeS_3_] (**1_2_**). In addition, the presence of [TiCp_2_S_5_] (**0_1_**) was postulated.

In the current contribution, the preliminary work was extended to the preparation of five solid solutions of mixed titanocene selenide sulfides [TiCp_2_Se*_x_*S_5−*x*_] (Cp = C_5_H_5_^−^) with the initial Se:S ranging from 1:4 to 4:1 (phases ***B***–***E***). Their ^77^Se- and ^13^C-NMR spectra were recorded from the CS_2_ solution. The definite assignment of the NMR spectra was based on the PBE0/def2-TZVPP calculation of the ^77^Se chemical shifts that verified the earlier, tentative inferences. Further confirmation was provided by ^13^C-NMR spectra of the samples. The presence of [TiCp_2_S_5_] (**0_1_**) and [TiCp_2_Se_5_] (**5_1_**) was unambiguously verified. By comparison of the relative intensities of the ^13^C NMR resonances with those of ^77^Se resonances, a complete, consistent assignment of the ^13^C spectra could be made.

All crystal structures of the mixed Se–S phases showed disorder in the chalcogen-atom positions. The site occupancy factors of selenium and sulfur in the solid phases ***B***–***E*** were consistent with classification of the phases as solid solutions of the species observed in the corresponding CS_2_ solutions even though the selenium contents of the solid phases ***B***–***E*** were higher than those in the corresponding solutions due to the rapidly decreasing solubility of the [TiCp_2_Se*_x_*S_5−*x*_] complexes, as the selenium content increases.

The enthalpies of formation were calculated for all twenty [TiCp_2_Se*_x_*S_5−*x*_] (*x* = 0–5) species at DLPNO-CCSD(T)/CBS level of theory and augmented to account for core-valence correlation and scalar relativistic, as well as spin-orbit coupling contributions to energies. The formation enthalpies could be divided into three distinct, but slightly overlapping groups. The most favourable group contained complexes with two Ti–S bonds, the intermediate group consisted of complexes with one Ti–S and one Ti–Se bond, and group with least exothermic enthalpy of formation contained complexes with two Ti–Se bonds. Within a given chemical composition, the isomers of the most favourable enthalpy of formation were those, which were observed by ^77^Se and ^13^C-NMR spectroscopy.

## Figures and Tables

**Figure 1 molecules-24-00319-f001:**
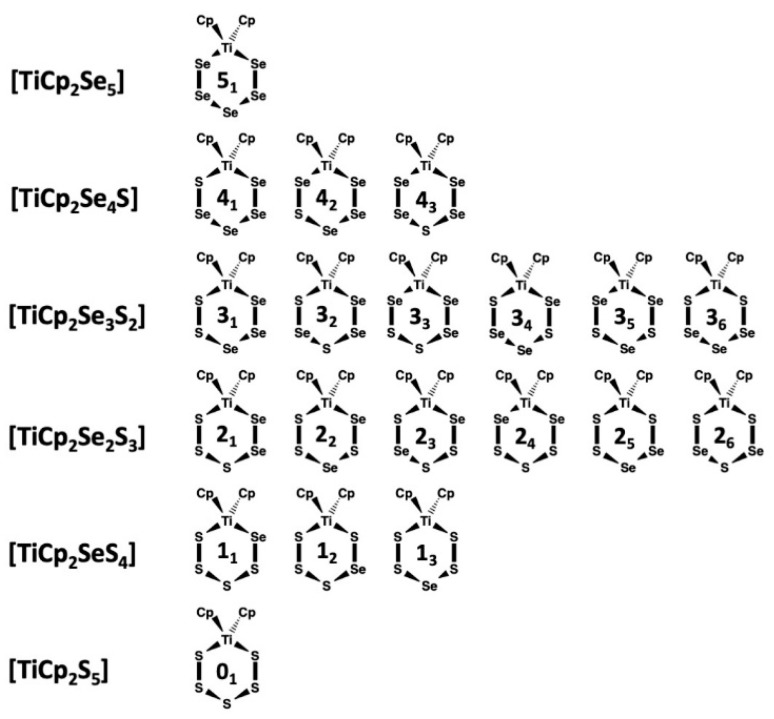
Possible isomers of [TiCp_2_Se*_x_*S_5−*x*_] (*x* = 0–5).

**Figure 2 molecules-24-00319-f002:**
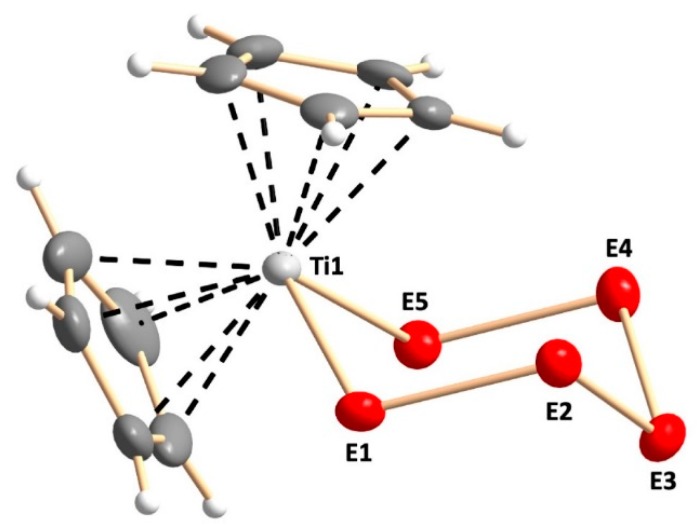
Molecular structure of [TiCp_2_Se*_x_*S_5−*x*_] as exemplified by Phase **C**. In phases ***B***–***E*** the chalcogen atom positions marked with “E*i*” (*i* = 1–5) are disordered with sulfur and selenium in random positions. The site occupancy factors are shown in Appendix A in the Appendix A. The anisotropic displacement parameters are shown at 50% probability level.

**Figure 3 molecules-24-00319-f003:**
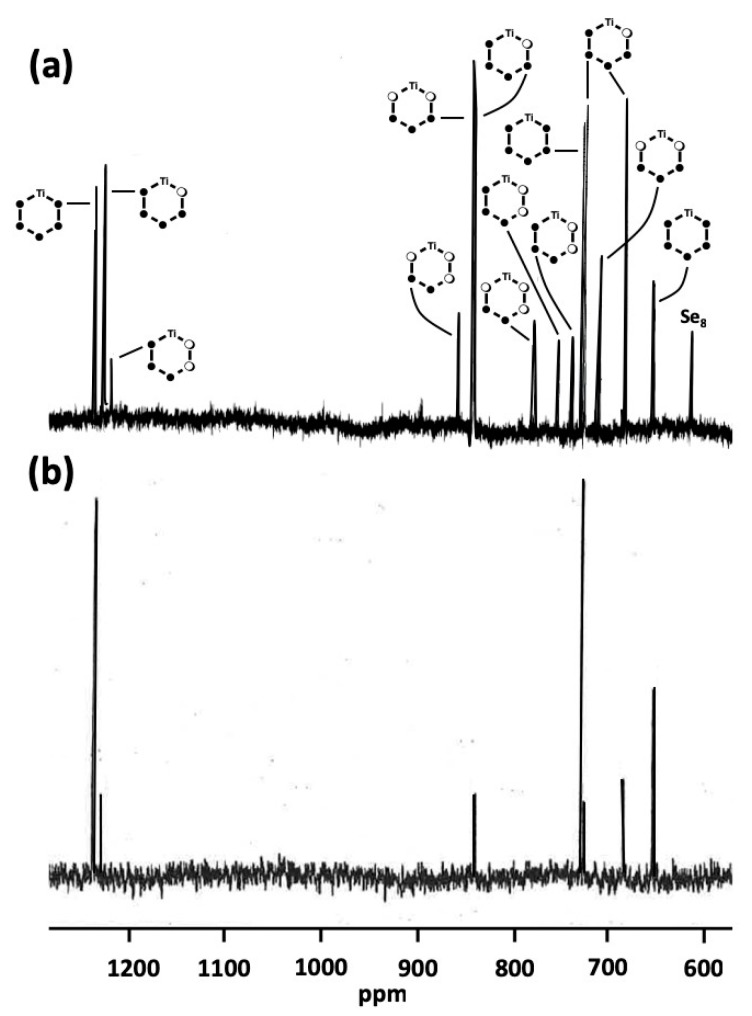
(**a**) The ^77^Se-NMR spectrum of phase ***D*** recorded previously in CS_2_ together with the tentative assignment [15] (adapted with permission from Pekonen, P.; Hiltunen, Y.; Laitinen, R.S.; Valkonen, J. Selenium-77 NMR Spectroscopic and X-ray Crystallographic Characterization of Bis(cyclopentadienyl)titanium Selenide Sulfide Mixtures [Ti(C_5_H_5_)_2_Se*_x_*S_5-*x*_]. *Inorg. Chem.*
**1991**, *30*, 1874–1878. Copyright 1991 American Chemical Society). (**b**) The ^77^Se-NMR spectrum of phase ***E*** recorded in CS_2_.

**Figure 4 molecules-24-00319-f004:**
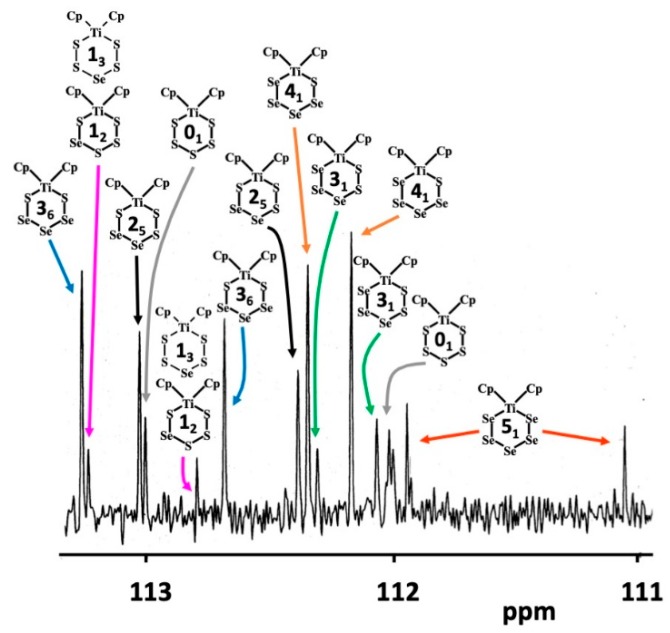
^13^C-NMR spectrum of phase ***D*** in CS_2_.

**Figure 5 molecules-24-00319-f005:**
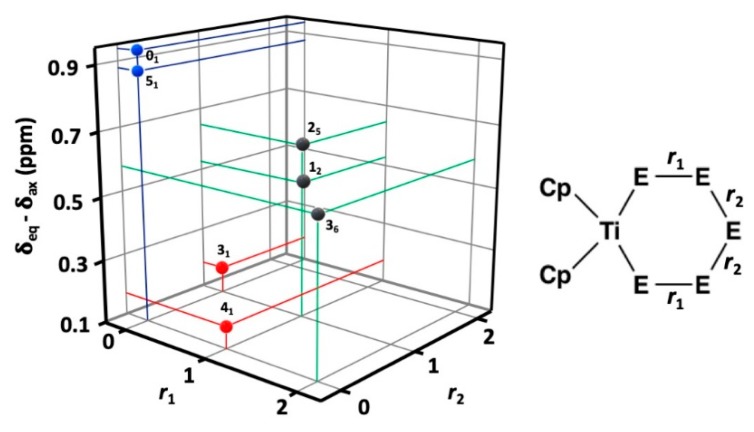
The difference between the ^13^C chemical shifts as a function of heteronuclear Se–S bonds in positions *r*_1_ and/or *r*_2_ (*r*_i_ = 0, 1, 2). The data points in blue mark complexes with no heteronuclear chalcogen-chalcogen bonds (**0_1_** and **5_1_**), those marked in red indicate complexes with one Se–S bond either in position *r*_1_ or *r*_2_, and those in grey indicate two Se–S bonds either in positions *r*_1_ or *r*_2_.

**Figure 6 molecules-24-00319-f006:**
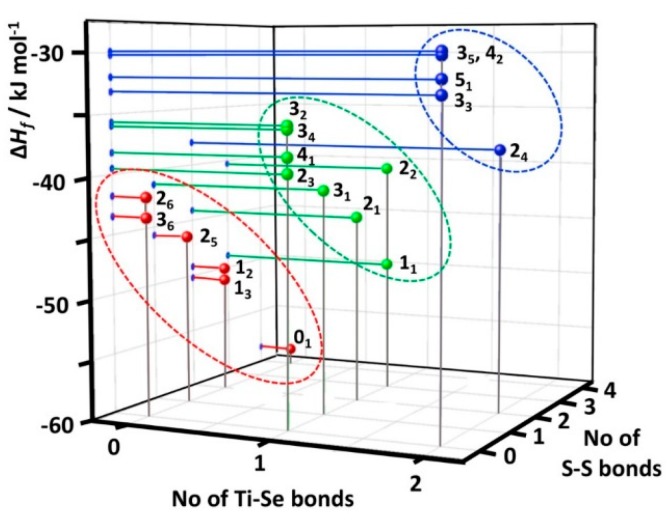
Enthalpies of formation of the [TiCp_2_Se*_x_*S_5−*x*_] complexes as a function of the number of Ti–Se and S–S bonds. Complexes with two Ti–Se bonds have been marked in blue, those with one Ti–S and one Ti–Se bond in green, and those with two Ti–Se bonds in red.

**Figure 7 molecules-24-00319-f007:**
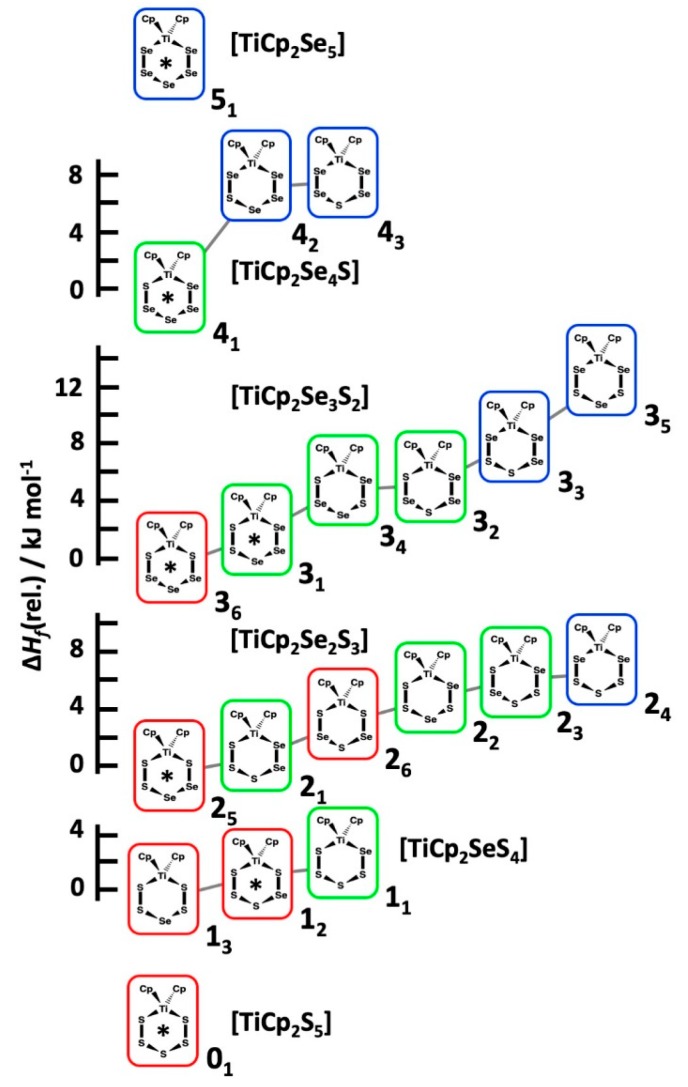
The relative enthalpy of formation of the [TiCp_2_Se*_x_*S_5−*x*_] isomers of the same composition. The most stable isomer is given the relative value of 0 for each composition. The red frames indicate two Ti–S bonds, green frames one Ti–S and one Ti–Se bond, and the blue frames two Ti–Se bonds. The complexes observed in the reaction mixtures by ^77^Se and ^13^C NMR spectroscopy have been indicated by an asterisk.

**Table 1 molecules-24-00319-t001:** Computed and observed chemical shifts of [TiCp_2_Se*_x_*S_5−*x*_] ^a^.

Complex	Intensity Ratio	E1	E2	E3	E4	E5
[TiCp_2_Se_5_] (**5_1_**) *Obs.*	2:2:1	1265 *1238*	730 *728*	696 *654*	730 *728*	1265 *1238*
[TiCp_2_Se_4_S] (**4_1_**) *Obs.*	1:1:1:1	1235 *1229*	723 *680*	728 *725*	832 *840*	
[TiCp_2_Se_3_SSe] (**4_2_**)	1:1:1:1	1248	750	729		1326
[TiCp_2_Se_2_SSe_2_] (**4_3_**)	1:1	1247	784		784	1247
[TiCp_2_Se_3_S_2_] (**3_1_**) *Obs.*	1:1:1	1228 *1221*	736 *737*	764 *752*		
[TiCp_2_Se_2_SSeS] (**3_2_**)	1:1:1	1214	779		875	
[TiCp_2_Se_2_S_2_Se] (**3_3_**)	1:1:1	1231	828			1300
[TiCp_2_SeSSe_2_S] (**3_4_**)	1:1:1	1296		761	852	
[TiCp_2_SeSSeSSe] (**3_5_**)	2:1	1311		753		1311
[TiCp_2_SSe_3_S] (**3_6_**) *Obs.*	2:1		825 *841*	745 *710*	825 *841*	
[TiCp_2_Se_2_S_3_] (**2_1_**)	1:1	1209	812			
[TiCp_2_SeSSeS_2_] (**2_2_**)	1:1	1285		790		
[TiCp_2_SeS_2_SeS] (**2_3_**)	1:1	1267			917	
[TiCp_2_SeS_3_Se] (**2_4_**)	1	1288				1288
[TiCp_2_SSe_2_S_2_] (**2_5_**) *Obs.*	1:1		850 *858*	789 *778*		
[TiCp_2_SSeSSeS] (**2_6_**)	1		870		870	
[TiCp_2_SeS_4_] (**1_1_**)	1	1255				
[TiCp_2_SSeS_3_] (**1_2_**) *Obs.*	1		916 *936*			
[TiCp_2_S_2_SeS_2_] (**1_3_**)	1		818			

^a^ The experimental ^77^Se NMR chemical shifts are shown in italicized type and grouped with the species, for which the experimental values exhibit the closest agreement.

**Table 2 molecules-24-00319-t002:** Semiquantitative determination of the contents (mol%) of the [TiCp_2_Se*_x_*S_5−*x*_] complexes in phases ***B***–***E*** based on the relative intensities observed in the ^77^Se and ^13^C NMR spectra of the reaction mixtures recorded in carbon disulfide.

Complex	^77^Se Chemical Shifts (ppm)					^13^C Chemical Shifts (ppm)				
		*B* ^a^	*C* ^a^	*D* ^a^	*E*		*B*	*C*	*D*	*E*
**5_1_**	1238, 728, 654 (2:2:1)	-	5	19	65	113.93, 111.06	-	5	19	65
**4_1_**	1229, 840, 725, 680 (1:1:1:1)	6	24	40	31	112.35, 112.17	8	27	41	35
**3_1_**	1221, 752, 737 (1:1:1)	-	6	4	4	112.31, 112.07	-	6	7	-
**3_6_**	841, 710 (2:1)	14	28	24	-	113.26, 112.68	15	27	21	-
**2_5_**	858, 778 (1:1)	20	16	6	-	113.03, 112.39	23	13	5	-
**1_2_**	936	15	6	-	-	113.24, 112.80	9	-	-	-
**0_1_** ^b^		45	15	7	-	113.01, 112.02	45	15	7	-

^a^ The relative contents of [TiCp_2_Se*_x_*S_5-*x*_] complexes observed in the ^77^Se-NMR spectra have been scaled to take [TiCp_2_S_5_] (**0_1_**) into account. ^b^ The content of [TiCp_2_S_5_] (**0_1_**) has been estimated from the ^13^C-NMR spectra.

**Table 3 molecules-24-00319-t003:** The composition of solid solutions ***B***–***E*** based on the observed disorder in the crystal structures (see Appendix A).

Complex	Phase *B*	Phase *C*	Phase *D* ^a^	Phase *E*
**5_1_**	-	12	25	65
**4_1_**	33	25	23	35
**3_1_**	-	9	4	-
**3_6_**	1	18	24	-
**2_5_**	23	21	6	-
**1_2_**	2	6	-	-
**0_1_**	41	9	18	-

*^a^* See Reference [15].

**Table 4 molecules-24-00319-t004:** Crystal data and details of structure determination of crystals from phases ***B*** (initial Se:S ratio 1:4) ***C*** (initial Se:S ratio 2:3), and ***E*** (initial Se:S ratio 4:1).

	Phase *B*	Phase *C*	Phase *E*
Empirical formula	C_10_H_10_S_3.17_Se_1.83_Ti	C_10_H_10_S_2.12_Se_2.88_Ti	C_10_H_10_S_0.35_Se_4.65_Ti
Formula weight	424.21	473.57	556.46
Temperature (K)	120	120	120
Crystal colour, habit	Dark red, Needle	Dark red, Block	Dark red, Block
Crystal dimensions (mm^2^)	0.450 × 0.150 × 0.100	0.180 × 0.120 × 0.080	0.200 × 0.150 × 0.120
Crystal system	Monoclinic	Monoclinic	Triclinic
*a* (Å)	13.000(3)	13.091(3)	8.011(2)
*b* (Å)	7.950(2)	8.062(2)	8.135(2)
*c* (Å)	14.300(3)	14.277(3)	11.791(2)
α (^o^)			96.46(3)
β (^o^)	114.20(3)	114.29(3)	105.84(3)
γ (^o^)			108.51(3)
*V* (Å^3^)	1348.0(6)	1373.4(6)	684.1(3)
Space Group	*P*2_1_/*c*	*P*2_1_/*c*	*P*-1
*Z*	4	4	2
*D*_calc_ (g/cm^3^)	2.090	2.290	2.702
*F*(000)	820	896	511
μ(MoKα) (cm^−1^)	6.035	8.553	13.019
No. of reflections measured	7403	6434	4558
No. of unique reflections	2337	2374	2264
No. of observed reflections/No. Variables	2109/151	2089/151	2077/148
Reflection/Parameter Ratio	13.97	13.83	14.03
Min. and Max. Transmissions			
*R* _INT_	0.0299	0.0523	0.1025
*R*_1_ [*I* > 2σ(*I*)] ^a^	0.0256	0.0345	0.0695
*R*_1_ (all reflections) ^a^	0.0299	0.0421	0.0733
*wR*_2_ (all reflections) ^b^	0.0638	0.0859	0.1840
Goodness of fit	1.086	1.192	1.054
Max., min. residual electron density (e^−^/Å^3^)	0.381, −0.297	0.756, −0.634	2.562, −1.829
CCDC No.	1887990	1887988	1887989

^a^*R*_1_ = Σ||*F_o_*|−|*F_c_*||/Σ|*F_o_*|. ^b^
*wR*_2_ = [Σ*w*(*F_o_*^2^ − *F_c_*^2^)^2^/Σ*wF_o_*^4^]^½^.

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
