# Peer review of "Titanocene Selenide Sulfides Revisited: Formation, Stabilities, and NMR Spectroscopic Properties"

_molecules, 2019, doi:10.3390/molecules24020319_

Reviewer 1 Report

The manuscript by Tiainen et al. presents a detailed study of titanocene mixed selenide-sulfide derivatives. A comprehensive analysis of reaction products of various initial Se:S ratios of the reactants by means of 77Se and 13C NMR spectroscopy, X-ray diffraction of the newly obtained phases, all supported by computational analysis of chemical shifts and relative enthalpies of formation. This work follows some previous investigations into this class of compounds and it provides a valuable contribution showing a meaningful utilization of various methods to assign complex mixtures of isomeric products.

I did not really find any reason for criticism. I am just listing several minor corrections recommended:

page 2, line 65 – “of all mixed the five phases” should probably be “of all mixed five phases” or “all the five phases”

page 5, line 123 – “it is not possible with to assign…” should be “it is not possible to assign…”

page 5, line 131 – “of the hexatatomic…” should be “hexaatomic” – please also include a reference to this statement (conformational chair-chair inversion of TeE5 ring…)

page 5, line 137 - …of the pairs 13C resonances…” should be “of the pairs of 13C resonances…”

Could the authors also comment on the possible role of solvent effects (especially when a more polar solvent is taken into account) on the enthalpies of formation of the molecules. Is there expected any significant change in the order of energies and thus in the predicted composition of the reaction mixtures?

In conclusion, I recommend the paper to be published as is (spellcheck is recommended).

Author Response

We are grateful to the reviewer for carefully reading the manuscript and correcting the typographical errors. All corrections have been marked in red in the revised manuscript.

The reviewer asked about the role of solvent in the calculations. Due to the similarity of sulfur and selenium, the solvent effects were virtually identical. the test computations for [TiCp2S5] and [TiCp2Se5] indicated that inclusion of the solvent effects by PCM stabilized both complexes by 12.9 and 12.8 kJ/mol, respectively. All [TiCp2SexS5-x] test species showed the stabilization of 12.7-12.9 kJ/mol. Solvent effect therefore do not change the relative stabilities of the complexes, and therefore, due to computational economy, only the vacuum calculations were performed for all complexes.

Reviewer 2 Report

In this paper, the authors reported the preparation of [TiCp2S5] (phase A), [TiCp2Se5] (phase F), and five solid solutions of mixed titanocene selenide sulfides [TiCp2SexS5-x] with the initial Se:S ranging from 1:4 to 4:1 (phases B-E), and the definite assignment of their NMR spectra by using the PBE0/def2-TZVPP calculations. The crystal structures of phases BC, and E were also determined. In addition, the relative stabilities of different [TiCp2SexS5-x] (x = 0-5) were calculated at DLPNO-CCSD(T)/def2-XZVPP (X = T, Q) level of theory. These results provide new information of the composition of the crystalline solid solutions of [TiCp2SexS5-x] phases, and therefore, the referee recommends that this paper is published in Molecules after the following revisions.

1) 3.1. Preparation of [TiCp2SexS5-x],

The experimental procedures should be described in more detail for the readers to reproduce the preparation of CS2 solutions of [TiCp2SexS5-x] for the 77Se and 13C NMR measurements and the recrystallization for the crystal structure determinations.

2) Lines 237-238,

“Phase A (Se:S = 1:4), phase (Se:S = 2:3), phase (Se:S = 3:2), and phase (Se:S = 4:1)” should be corrected to “Phase B (Se:S = 1:4), phase (Se:S = 2:3), phase (Se:S = 3:2), and phase (Se:S = 4:1)”.

3) Line 249,

Please check the temperature value (150 K). This value is different from that described in Table 4.

Author Response

(1) The synthetic details have been added in Supplementary Material (Section 4).

(2) The designation of the phases have been corrected (marked in red in the manuscript).

Reviewer 3 Report

The authors reported the synthesis, structure analysis, and NMR spectroscopic properties of titanocene selenide sulfides in this manuscript. Moreover, it is also important to estimate the stabilities of each titanocene selenide sulfides by quantum chemical calculations using a domain-based local pair-natural orbital coupled cluster (DLPNO-CCSD(T)) approximations which is accurate and cost-efficient methodology for the estimation of the enthalpies of formation for organic compounds. The work seems to have been conducted to a high standard and will appeal to a wide range of organometallic chemists, as such I would recommend acceptance of the manuscript after addressing a few minor issues.

P1. L16: Change “Cp = C5H4-“ to “Cp = C5H5-“

P3. L96: Change “2.2. 77. Se” to “2.2. 77Se”

P5. L131: Change “TeE5” to “TiE5”

P6. L142: Change “[TiCp2Se4S]” to “[TiCp2Se3S2]”

P9, L210: Put a period at the end of the sentence. Put after (see Figure 7)?

P11, L237: Change “Phase A, phase B, phase, C, phase D” to “Phase B, phase C, phase, D, phase E”

P11, Table 4: Crystal colour cannot find in table 4 though there is “crystal colour, habit” in this table. Please add the color information of crystals.

P13, L317: Change “Cp = C5H4-“ to “Cp = C5H5-“

P14, L359: There are two periods after the author.

P14, L365-371: Change “(11) (12) (13)” to “[11] [12] [13]”

P14, L379: Remove “ Sulfur Compounds”

P14, L380: Change “Phosphurys” to “Phosphorus”

P14, L389, 390: Change “TI” to “Ti”

P14, L397: Please italicize “Chem. Ber.”

P15, L434: Change “pf” to “of”

References [35] and [62], [36] and [68] are duplicated, respectively. Please revise it.

Author Response

We are grateful to the reviewer for carefully reading the manuscript and correcting the typographical errors. We have also removed the duplicated references and renumbered them. All corrections have been marked in red in the revised manuscript